



# How observations from automatic hail sensors in Switzerland shed light on local hailfall duration and compare with hailpads measurements

Jérôme Kopp[1], Agostino Manzato[2], Alessandro Hering[3], Urs Germann[3], and Olivia Martius[1]

[1]Oeschger Centre for Climate Change Research and Institute of Geography, University of Bern, Bern, Switzerland
[2]ARPA FVG - OSMER, Palmanova, Italy
[3]Federal Office of Meteorology and Climatology MeteoSwiss, Locarno-Monti, Switzerland

**Correspondence:** Jérôme Kopp (jerome.kopp@giub.unibe.ch)

**Abstract.** Measuring hailstorms is a difficult task due to the rarity and mainly small spatial extent of the events. Especially, hail observations from ground-based time-recording instruments are scarce. We present the first study of extended field observations made by a network of 80 automatic hail sensors from Switzerland. The main benefits of the sensors are the live recording of the hailstone kinetic energy and the precise timing of the impacts. Its potential limitations include a diameter dependent dead time which results in less than 5% of missed impacts, and the possible recording of impacts not due to hail which can be filtered using a radar reflectivity filter. We assess the robustness of the sensors measurements by doing a statistical comparison of the sensor observations with hailpads observations and we show that despite their different measurement approaches, both devices measure the same hail size distributions. We then use the timing information to measure the local duration of hail events, the cumulative time distribution of impacts and the time of the largest hailstone during a hail event. We find that 75% of local hailfalls last just a few minutes (from less than 4.4 min to less than 7.7 min, depending on a parameter to delineate the events) and that 75% of impacts occurs in less than 3.3 min to less than 4.7 min. This time distribution suggests that most hailstones, including the largest, fall during a first phase of high hailstone density, while a few remaining and smaller hailstones fall in a second low density phase.

## 1 Introduction

Measuring hailstorms is a difficult task due to the rarity and mainly small spatial extent of the events. Hail typically happens less than one time per year at any location in Europe (Punge and Kunz, 2016) and around 2-3 times per square kilometer per year in areas that are considered to be prone to hailstorms in Switzerland (Nisi et al., 2016; NCCS, 2021). The need for reliable, high quality, long-term observational data for hail has been repeatedly highlighted by the hail community in recent years (Punge and Kunz, 2016; Martius et al., 2018; Allen et al., 2020).

The two main approaches to observe and measure hail are (1) using proxy data obtained from remote sensing instruments, particularly weather radars or satellites, and (2) using surface (or ground-truth) observations. Surface observations can be obtained from different sources: crowdsourcing mobile applications, such as the MeteoSwiss app (Barras et al., 2019); insur-





ance damage claims from insurance companies; observations from storm chasers (eg. sturmarchiv.ch) or observers networks (Changnon, 1970; Počakal et al., 2009; Nađ et al., 2021); observations from aerial drone measurements (Soderholm et al.,
2020, Lainer et al. ?) and hailpad networks (Changnon, 1970; Federer et al., 1986; Smith and Waldvogel, 1989; Fraile et al., 2003; Giaiotti et al., 2003; Sánchez et al., 2009; Pocakal, 2011; Manzato, 2012).

Among those ground-based observational methods, hailpad networks have been the most extensively used. Hailpads are cheap extruded polystyrene foam rectangles that are exposed outdoors (Towery et al., 1976). Upon impact, the hailstone leaves a dent in the hailpad, whose size depends on the hailstone dimension. Hailpads are manually collected and replaced by
volunteers after each hailstorm. While the collection date is always recorded, hailpads provide time-integrated measurements and consequently don't give any information about the precise start and end of a hailfall, nor the exact timing of each single hailstone impact.

Observations from ground-based time-recording instruments for hail documented in the literature are limited. Federer and Waldvogel (1975) observed a single hailstorm in Switzerland using a hail spectrometer, where hailstones fall on a surface,
are then photographed with an automatic camera, and removed before the cycle starts again. Brown et al. (2014) recorded three datasets in the Great Plains region of the United States using an impact disdrometer. Giammanco et al. (2016) collected data from four thunderstorms during a field campaign in 2015 using a network of six hail impact disdrometers. Consequently, there is few literature discussing local hailfall duration and time evolution of the hail size distribution, which is important for understanding hail, constraining hail parametrization schemes in numerical models, and for validating radar-based hail
algorithms.

Switzerland completed the installation of the first national scale network of time-recording instruments for hail in 2020, composed of 80 automatic hail sensors. The automatic hail sensors that comprise the network (Wetzel, 2018) are based on a prototype presented by Löffler-Mang et al. (2011). The observational dataset now consists of about 12'300 hailstone impacts. Some observations recorded during the particularly active hail season of 2021 were presented in Kopp et al. (2022). However,
a comprehensive analysis of the full observational dataset, as well as an in-depth discussion of the capabilities of the automatic hail sensors are still missing.

The objective of this paper is to present the first study of extended field observations made by automatic hail time-recording instruments. More specifically, we address the following questions:

– What are the key operational aspects of automatic hail sensors? What measurements do the sensors provide? Considering
the technical aspects, how to make best use of the sensor observations and what new information can they provide about hail?

– How do sensor observations compare with hailpad observations? What can we learn from this comparison?

– What is the point (local) duration of hailfall in Switzerland ? How does it compare with existing literature estimates?

– How are hailstone impacts distributed in time during a hailfall?



We present the hail sensor and its measurement process with its advantages and potential shortcomings in section 2.1. We show examples of time series of hailstone impacts captured by the sensors to illustrate our methodology to characterize a local hail event in 2.2. We introduce the hailpad data used for comparison in section 2.3. Section 3.1 presents general observations of the network. Those observations are subsequently compared with those of a hailpad network from northern Italy (Manzato et al., 2022) in section 3.2.1. Section 3.3 presents the results of the analysis of time-related quantities such as the local hailfall

duration 3.3.1, the cumulative time distribution of impacts 3.3.2 and the time of occurrence of the largest hailstone 3.3.3. Finally, general conclusions and future research avenues are presented in section 4.



## 2 Data and Methods

### 2.1 Automatic hail sensors

#### 2.1.1 The network

In the "Swiss Hail Network" project, 80 automatic hail sensors were installed between June 2018 and July 2020 in the three most hail-prone regions of Switzerland according to the climatology (Nisi et al., 2016, 2018; NCCS, 2021): the Jura (15 sensors) and Napf (38 sensors) north of the Alps and Southern Ticino (27 sensors) south of the Alps (Fig. 1). The main purpose of the Swiss Hail Network is to collect ground observations of hail from several events that can then be used to a) verify operational radar-based hail algorithms and hail information from hailpads, and b) for scientific studies on hail in general. This project is a public-private partnership between La Mobilière, MeteoSwiss, inNET Monitoring AG and the University of Bern. The sensors will operate for at least 8 years and provide near real-time data on hailstorms. As of Spring 2023, the sensors have now been operating between three to five hail seasons (April to September) depending on their location.

#### 2.1.2 Measurement process

Each sensor is designed as a Makrolon thermoplastic disc with a diameter of $50 \, \text{cm}$ (Fig. 2), providing a sensing area of approximately $0.196 \, m^2$. The disc oscillates when hit by a hailstone, and a highly sensitive piezoelectric microphone records the oscillations, which are then converted to the hailstone kinetic energy (Joule) through a log-linear calibration curve.

The calibration procedure (Riehle and Schön, 2021), which allows to convert the electric signal output to an estimate of the kinetic energy, is a key step of the measurement process. Each sensor is individually calibrated by the manufacturer under laboratory conditions before its delivery (lab calibration). As each sensor is exposed to various weather conditions throughout the year, it has to be re-calibrated once a year before or at the beginning of the hail season (field calibration). The field calibration is done using a portable calibration unit that can be fixed to the sensor. Three masses of different known weights are each dropped twelve times from two fixed heights. The average of the signal responses is calculated for each of the six different mass-height combinations, giving six points used to fit a power law between the voltage signal and the kinetic energy. This power law is then used as the calibration curve to translate the voltage signal of hailstone impacts to a kinetic energy estimate.

#### 2.1.3 Hailstone diameter estimation

The hailstone diameter is then determined from the kinetic energy, assuming spherical hailstones with constant drag coefficient (Pruppacher and Klett, 2010):

$$D = \sqrt[4]{\frac{9 \cdot E_{kin} \cdot \rho_{air} \cdot c_w}{\rho_{ice}^2 \cdot \pi \cdot g}} \qquad (1)$$

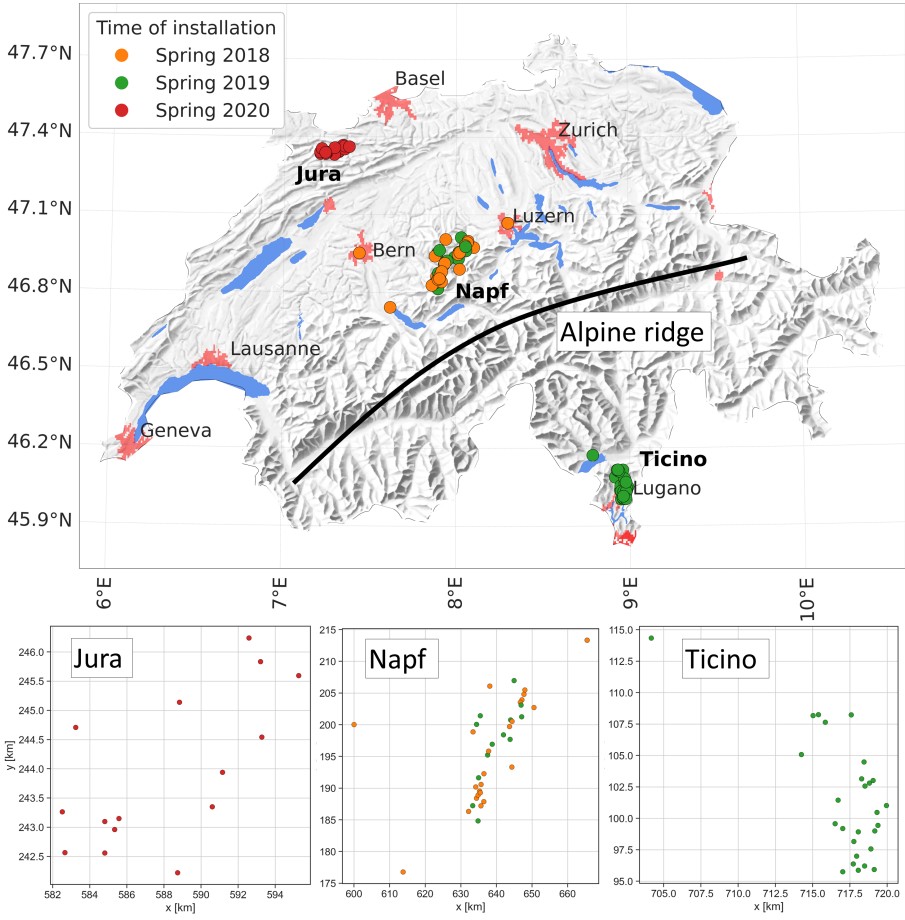

**Figure 1.** Top: Map of Switzerland showing the locations of the 80 sensors according to their installation date in the three hail-prone regions (Jura: 15, Napf: 38, Ticino: 27); red patches show urban areas and the black line denotes the alpine ridge. Bottom: Zoom-in of the three hail-prone regions showing network density, scale in kilometers.

where $D$ is the equivalent spherical hailstone diameter, $E_{kin}$ is the kinetic energy of the hailstone, $\rho_{air} = 1.2 \frac{kg}{m^3}$ is the surrounding air density, $c_w = 0.5$ is the drag coefficient, $\rho_{ice} = 870 \frac{kg}{m^3}$ is the hailstone ice density, $g = 9.81 \frac{m}{s^2}$ is gravity. Diameter calculations using Eq. 1 is directly implemented in the hail sensor software.

While Eq. 1 has been successfully used in early literature (eg. Federer and Waldvogel, 1975; Ulbrich and Atlas, 1982), more recent literature has shown that the assumptions on which it is based are not always satisfied. First, hailstone growth results in a variety of hailstone shapes, and hailstones tend to become increasingly non-spherical with increasing size (see for example Shedd et al., 2021). Then, the drag coefficient of hailstones (even spherical ones, but to a lesser extent) depends on the Reynolds number and their density can vary greatly (see for example Heymsfield et al., 2014, 2018, 2020).





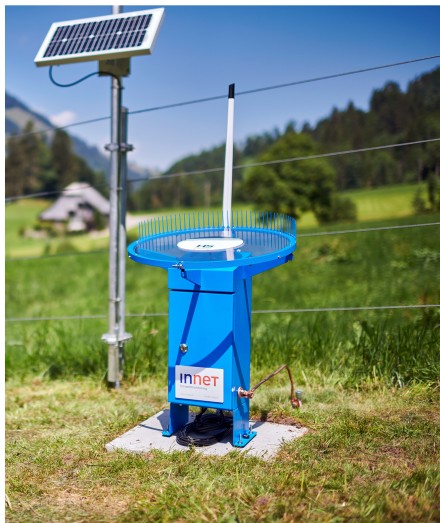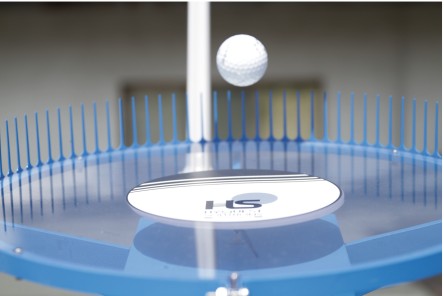

**Figure 2.** Left: Picture of one of the automatic hail sensors installed in the project "Swiss Hail Network" (© Manu Friederich). Right: Zoom on the sensor Makrolon disc as a golf ball is falling (La Mobilière / Sascha Moetsch)

We focus on relatively small hailstones, most of them with an estimated diameter of less than 20 mm. Therefore the assumption of spherical hailstones remains a rather good approximation and the effect on the drag coefficient is limited. However, we note that as the hail sensor primary output is the hailstone kinetic energy, other relations than Eq. 1 could be used and compared to estimate the equivalent hailstone diameter.

### 2.1.4 Known sources of uncertainties

As the sensor is continuously exposed to variable weather conditions, it is likely that its sensitivity slightly changes in the course of the year. The ambient temperature also influences the calibration process (personal communication from inNET AG). Thus, despite the yearly field calibration, the sensitivity of the sensor to weather conditions introduces uncertainty in the kinetic energy measurements. Another source of uncertainty is the impact location on the sensor plate. The piezoelectric microphone is located under the center of the Makrolon disc and consequently an impact close to the border of the disc will result in a slightly lower signal. The manufacturer indicates a 20% uncertainty in the estimation of the kinetic energy and recommends to work with hail classes of 5 mm diameter ranges for analysing a single hailstorm, although the sensor produces measurements with an accuracy of several decimal places (Riehle and Schön, 2021).

### 2.1.5 Sensor dead time and saturation

A known and necessary limitation of the automatic hail sensor is the "dead time", i.e., the time period following each impact during which no other hailstone can be recorded. The dead time allows the sensor to properly record an impact by avoiding interference from other hailstones hitting the sensor right after this first impact and by letting the sensor electronics perform





the necessary signal treatment. The dead time of the sensor ranges from 64 milliseconds for hailstones smaller than 10 mm to nearly 1 second for hailstones of about 35 mm (personal communication from the sensor manufacturer), which is the size range of the largest hailstone observed so far by the network. A dead time of 64 milliseconds corresponds to 15 impacts/second on the sensor plate or 70 impacts/second per $m^2$. When the sensor is not able to record a new impact because it happens during the dead time of a previous impact, we call it saturation.

We investigated the influence of saturation and quantified to which extent it affects the measured hailstones density (number of hailstones per second). We used an approach from radiation detection (Lucke, 1976) to estimate the "true" detection rate R:

$$R = \frac{N}{\left(T - \sum_{i=1}^{N} \tau_i\right)} \tag{2}$$

where $N$ is the number of recorded impacts, $T$ is the duration of a hail event and $\tau_i$ is the dead time of the $i^{th}$ hailstone. Equation 2 has been adapted to account for the hailstone size dependence of the dead time (hence the $\tau_i$'s). We then multiply $T$ by $R$ to obtain an adjusted number of impacts and we use this adjusted number to estimate the fraction of missed impacts.

$T$ (and subsequently $R$) depends on how we characterize and define a hail event (see sections 2.2 and 2.2.1 for details). At this stage, it is sufficient to say that the definition depends on one parameter, called the maximum blank time or $t_{mb}$, and that the estimated fraction of missed impacts, averaged over all hail events, takes values between 4% and 4.6% for the considered values of $t_{mb}$. Hence, the average fraction of missed impacts remains low, compared to all impacts. The fraction can be higher for individual events (up to 10%), especially those events with a higher hit rate (hailstones per second). We also note that we cannot know the diameters of the missed hailstones.

Finally, hailpads can also become saturated (Manzato et al., 2022). Saturation on hailpads can happen when its surface become covered of dents (impacts) made by numerous hailstones, such that subsequent hailstones can fall inside those existing dents and cannot be distinguished. For a more in-depth discussion about hailpad saturation, we refer the interested reader to Manzato et al. (2022).

### 2.1.6 Minimum signal threshold

A minimum signal threshold is set by the manufacturer after the lab calibration to avoid recording impacts of large rain drops or graupel, whose kinetic energies could approach those of small hailstones (Pruppacher and Klett, 2010). This threshold is initially set to correspond to a 5 mm diameter hailstone, consistent with the WMO definition of hail, that is: precipitation of ice particles with a diameter larger than 5 mm (World Meterological Association, 2017).

However, a close examination of the sensor data revealed that this threshold had not been adjusted after the field calibrations. Consequently, it happened that in some cases the threshold no longer exactly corresponded to a 5 mm hailstone but to a larger or smaller diameter according to the new field calibration.

In the case of a less than 5 mm threshold, this led to the recordings of graupel or possibly large rain drops. For such cases, a simple filtering of all impacts with an estimated diameter lower than 5 mm can correct the data.





In the case of a more than 5 mm threshold, some small hailstones larger than 5 mm might not have been recorded properly. This is more problematic as the exact number of missed hailstones due to this higher threshold cannot be estimated.

It was not feasible to calculate the number of cases for which the threshold departed significantly from 5 mm from the archived calibration records. Considering the daily records by sensor, an examination of the data revealed that in 37 cases among 1'447 (2.5%), hailstones of less than 4 mm have been recorded. We cannot make the same estimation for threshold larger than 5 mm as it's not possible to say if this was indeed due to the threshold or because the hailstones were all larger. A reasonable assumption would be that cases where the threshold was significantly larger than 5 mm happened on average as often as cases where it was significantly smaller, leading to a 2.5% of cases where the threshold prevented the measurements of hailstones smaller than 6 mm. Although we have to bear in mind that in some cases the lower end of the hail size distribution had been truncated due to this larger than 5 mm threshold, we believe that a 2.5% missing rate is acceptable. We note that from the 2023 hail season onward the signal threshold will be adjusted after each field calibration such that the lower bound of diameters is fixed to 5 mm for all sensors and at all times.

### 2.1.7 Radar reflectivity filter

Not only hydrometeors can generate impacts on the sensor. Examples include animals touching the sensor such as birds, goats, cats and dogs, or flying objects in case of strong winds such as small branches or light gravel. For this reason, we use a radar reflectivity filter to ensure that there is a storm environment in close vicinity of the sensor. For that, we demand that the maximum reflectivity within a radius of 4km around the sensor at the time the sensor is hit is equal to or higher than 35 dBZ. This reflectivity threshold is operationally used in the thunderstorms radar tracking (TRT) algorithm by MeteoSwiss to identify storm-objects (Hering et al., 2004; Nisi et al., 2018). The 4km radius accounts for the wind drift of hailstones Barras et al. (2019). Impacts where the corresponding filter is not satisfied are not considered in this study.



## 2.2 Examples of measurements and event delineation

An interesting feature of the automatic hail sensors is that it provides the precise timing of each hailstone impact. This time information can be used to define the local duration of a hail event, just by looking at the first and last hailstones that hit the sensor and define those times as the beginning and end of the event. However, it is sometimes not possible to unambiguously define the beginning and end of an event.

To illustrate this, we show five examples of time series of impacts (Fig. 3. The event can be clearly identified in Fig. 3a. Almost 300 impacts are registered within 3 minutes with a clear start and end. The time series in Fig. 3b shows two impact clusters separated by 15 minutes, followed by 2 other impacts 45 minutes later. It is not straightforward to say whether the two first impact clusters come from the same hailstreak or from two distinct ones, due to the variability of the hailstreaks dimensions and storm velocities. Studies from the United States found that the majority of hailstreaks are less than 5 km wide, increasing to 10 to 15 km for more organized hailstorms, with maximum widths ranging from 25 to 30 km (Brimelow, 2018). Nisi et al. (2018) analysed a 15-year radar-based climatology of hailstreaks from Switzerland and found average streak lengths from 10.4 km to 36.2 km and average streak duration from 25 to 60 minutes. This gives average storm speed from 25 to 33 km/h. Trefalt et al. (2018) analysed a severe hailstorm in Switzerland and found that the storm mean velocity was approx. 6 km/h. They also found that other severe storms had speed ranging from 4.5 to 18.6 km/h. Using an object-based analysis of simulated thunderstorms in Switzerland, Raupach et al. (2021) found storm velocities ranging from a few km/h to 40 km/h. Combining those estimates of hailstreak areas and storm velocities, we conclude that the same storm can produce hail in the same place for durations ranging from a few minutes to just over an hour. This does not mean that hailstones would be produced continuously but that two or more series of hailstones separated by several minutes without hail (a "blank" period) could potentially be produced by the same storm at the same place.

Figure 3c,d,e present other examples of situations where hailstones impacts are separated by blank periods. 3e is particularly striking as only 18 impacts are registered in nearly 2 hours. One might ask the question if those impacts really correspond to hail. The average maximum reflectivity during the impacts is 47 dBZ and several neighbouring sensors also recorded impacts during the same time period with the same "scarce" pattern, indicating that hail was indeed responsible of the impacts. A more detailed analysis using radar data, numerical model data and crowdsourced observations would be needed to understand the storm patterns and attributing the impacts to distinct hailstreaks. As we cannot investigate all time series of impacts in such details, we need a simple conceptual model to group hailstone impacts into distinct events to further analyse hail duration.

### 2.2.1 Methodology for defining a hail event

We define a "hail event" as a series of consecutive impacts recorded by an individual sensor separated by less than time $t_{mb}$. If two impacts are separated in time by more than $t_{mb}$ then they belong to two different events. The choice of $t_{mb}$ determines the number of events and their duration. Large values of $t_{mb}$ will merge events, thereby decreasing the number of events and increasing their average duration. As an example, the time series of Fig. 3e is considered as a unique event for $t_{mb} = 20$



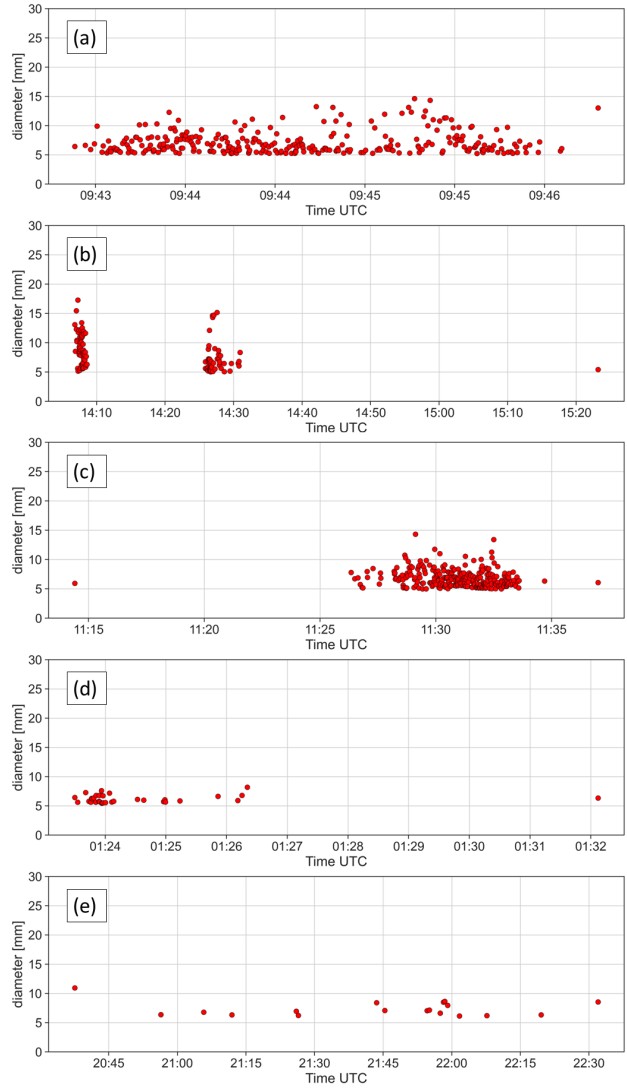

**Figure 3.** Time series of hailstones impacts, the y axis shows the estimated diameter (mm). Note the different time scales of the x axis. (a) July 8, 2021, Bironico Trafohaus, (b) July 24, 2021, Marbach Moesli, (c) July 26, 2021, Marbach Bergstation, (d) August 2, 2019, Lugano Onecar, (e) August 6, 2021, Cadro Schule.

minutes, while it is split in 6 distinct events for $t_{mb} = 10$ minutes: 3 events with a single impacts, 1 with 2 impacts, 1 with 3 impacts and another with 10 impacts.

We systematically considered values of $t_{mb}$ ranging from 1 minute to 2 hours and found that when $t_{mb}$ increases from 20 to 30 minutes, there is a jump in the increase of the average event duration (not shown). This suggests that we are grouping together events, which are separated by a longer period without hail. Blank periods of 30 minutes or more would imply both particularly large and slow moving storms. We also found that values of $t_{mb}$ of less than 5 minutes are too small and lead to an





artificially high number of events with a few impacts. However, as the choice of $t_{mb}$ can impact the subsequent analysis of the

local hailfall duration, we decided to investigate $t_{mb}$ values of 5, 10, 15 and 20 minutes more closely and present a sensitivity

analysis in the result section.



## 2.3 Hailpad data

The closest measurement device to an automatic hail sensor is a hailpad: they both measure hail at the ground on a surface of similar scale. Hailpads have an area of about 0.115 $m^2$, which is half the sensor area (0.196 $m^2$).

Hailpad studies covering different regions of the world were reviewed (Smith and Waldvogel, 1989; Fraile et al., 2003; Sánchez et al., 2009; Pocakal, 2011; Eccel et al., 2012). However, most studies provide only averaged quantities (number of impacts per hailpad or $m^2$, hit rate, etc.) and do not give much details on hailpad selection (e.g., is there a minimum number of dents below which hailpads are not considered?).

We work with hailpad observations from a stations network of NE Italy, collected during the 1988-2016 (29 years) warm seasons (Manzato et al., 2022), allowing for a more detailed comparison. The selection criteria applied to the hailpads were to keep only valid dents corresponding to hailstones of at least 6 mm and having an aspect ratio (major/minor axis) between 1 and 2. The reason was that dents corresponding to small diameters and very high aspect ratio likely do not represent true hailstone signatures. All hailpads with at least one valid impact were retained. This corresponds to 7'782 hailpads totalling 747'759 valid impacts. For details on the processing of hailpads and selection criteria, the interested reader is refer to Manzato et al. (2022).

The hailpads are usually collected and replaced by volunteers after each hailstorm. It is likely that the volunteer will wait for the storm activity to end before going outside to collect the hailpad. We also note that cases where the same station collected more than one hailpad in the same day are extremely rare, such that we can consider that each collected hailpad contains a daily aggregation of hail dents. Consequently, we also make daily aggregations of the impacts recorded by a given sensor and consider only impacts with estimated diameters of at least 6 mm to compare our data with the hailpad data in section 3.2.1. With these selections criteria, our sample is composed of 8'958 hailstones and 1'058 daily aggregations.





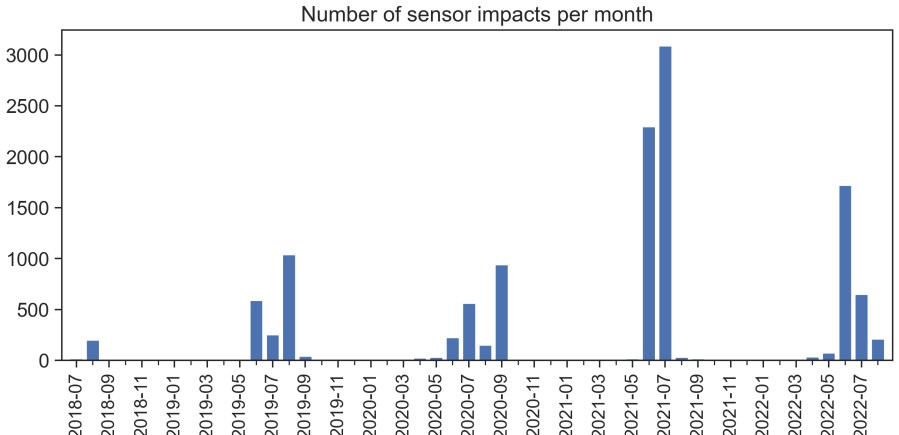

**Figure 4.** Monthly number of impacts (all sensors)

## 3    Results and Discussion

### 3.1    General observations, hailstones size and kinetic energy distributions

From July 2018 (when the first sensors were installed) to August 2022, 12'300 hailstone impacts were registered (Fig. 4). Few impacts were recorded in 2018 and 2019 as the sensor network was not yet fully deployed. The highest number of yearly

impacts was recorded in summer 2021, during a particularly active hail season (Kopp et al., 2022). The largest daily number of impacts for an event is 405 and has been recorded by the sensor of the Bergstation in Marbach (Napf region) on July 26, 2021 (see Fig. 3c).





Figure 5a shows the hailstone diameter probability distribution and 5b the kinetic energy distributions. The largest hailstone had an estimated diameter of 33 mm, which corresponds to a kinetic energy of 5.2 J. The median diameter is 6.7 mm and the

235 median kinetic energy is $9 \cdot 10^{-3}$ J. Only 41 impacts (0.33%) had an estimated diameter of 20 mm or more. An exponential fit (red line on Fig. 5a) of the hailstone probability distribution gives (with D in mm):

$$P(D) = 2.48(\pm 0.42) \cdot \exp\left(-0.37(\pm 0.03) \cdot D\right) \tag{3}$$

The fit works reasonably well for diameter up to 25 mm but underestimates the largest hailstone probabilities.

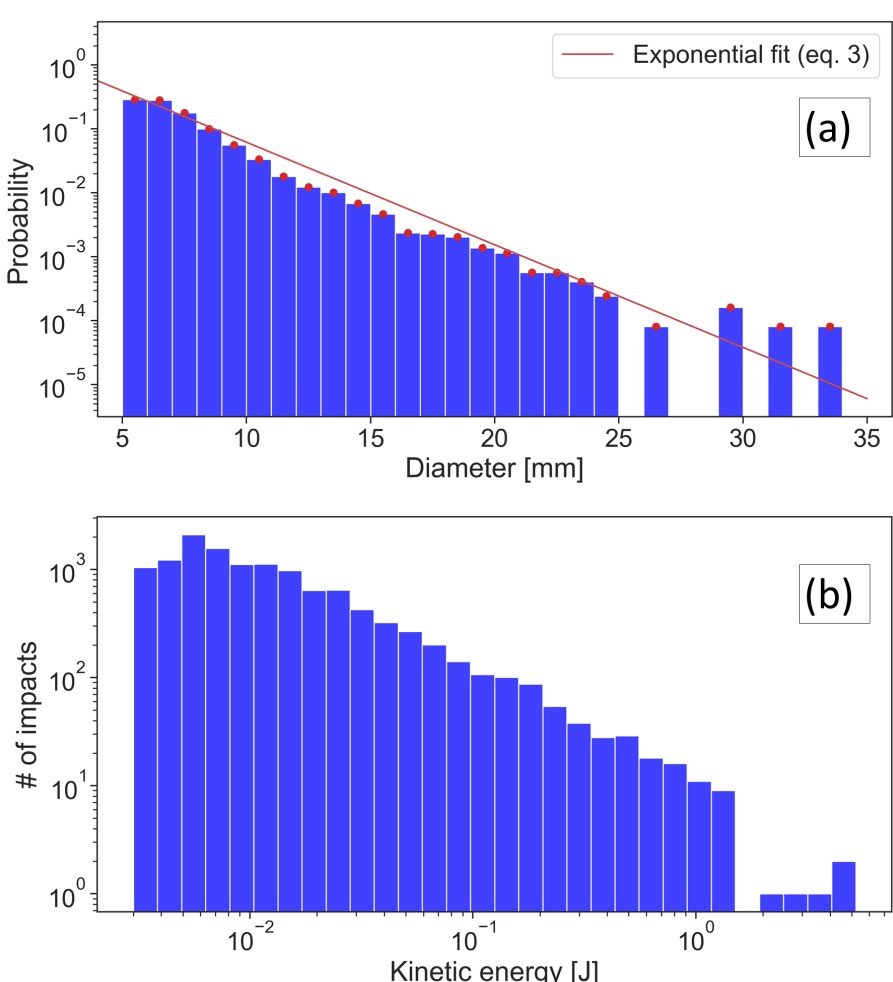

**Figure 5.** (a) Hailstone diameter probability distribution in 1 mm size classes (12'300 hailstones with diameter > 5 mm) with an exponential fit (red line). (b) Hailstone kinetic energy distribution, note the x log scale (same sample as in (a)).





## 3.2 Comparison with hailpads data

We first compare the distribution of the number of hailstones per hail sensors and and hailpads. Then we look at the averaged hail size distribution (HSD) at the ground by merging all measurements for each device.

### 3.2.1 Hail sensors and hailpads distributions

Figure 6 shows the probability distribution of the number of hailstones per hailpads and hail sensors. The two distributions differ substantially. There are proportionally much more sensors than hailpads with few (1 to 5) hailstones and fewer sensors than hailpads with more than 5 hailstones.

The difference in the distributions for large hailstones counts can be explained by the limited sample size of the sensors compared to the hailpads. Hence the tail of the distribution for hail sensors is probably missing large hailstone counts. Indeed, the largest number of hailstones registered by a hailpad from Manzato et al. (2022) is 1244, and 362 hailpads recorded a higher hailstone number than the largest number recorded by a sensor (405 hailstones).

Despite our smaller sample size, there are 507 sensor and only 193 hailpads with a single hailstone. A plausible explanation is that more than one hailfall is overlaid on the same hailpad. This could happen because volunteers checking hailpads with a low number of dents (possibly of small size) might not note them and would leave the hailpad exposed instead of replacing it with a new one. Volunteers could also not notice hailfalls with very low hailstone densities and not checking the hailpad at all. This would lead to a lower relative number of hailpads with a few hailstones and a higher relative number of hailpads with many hailstones. It might also be that despite the radar reflectivity filter, some impacts registered by the sensor were not caused by hailstones. A few exceptionally large rain drops or small wind-blown objects could generate those 1 to 5 impacts.

As stated in section 2.3 the area covered by a hailpad and a hail sensor are not the same. We looked at the distributions of the areal densities (impacts per $m^2$) by normalising both devices by their respective size. The same differences were noted in the distributions of the areal densities as in Fig. 6 (not shown).

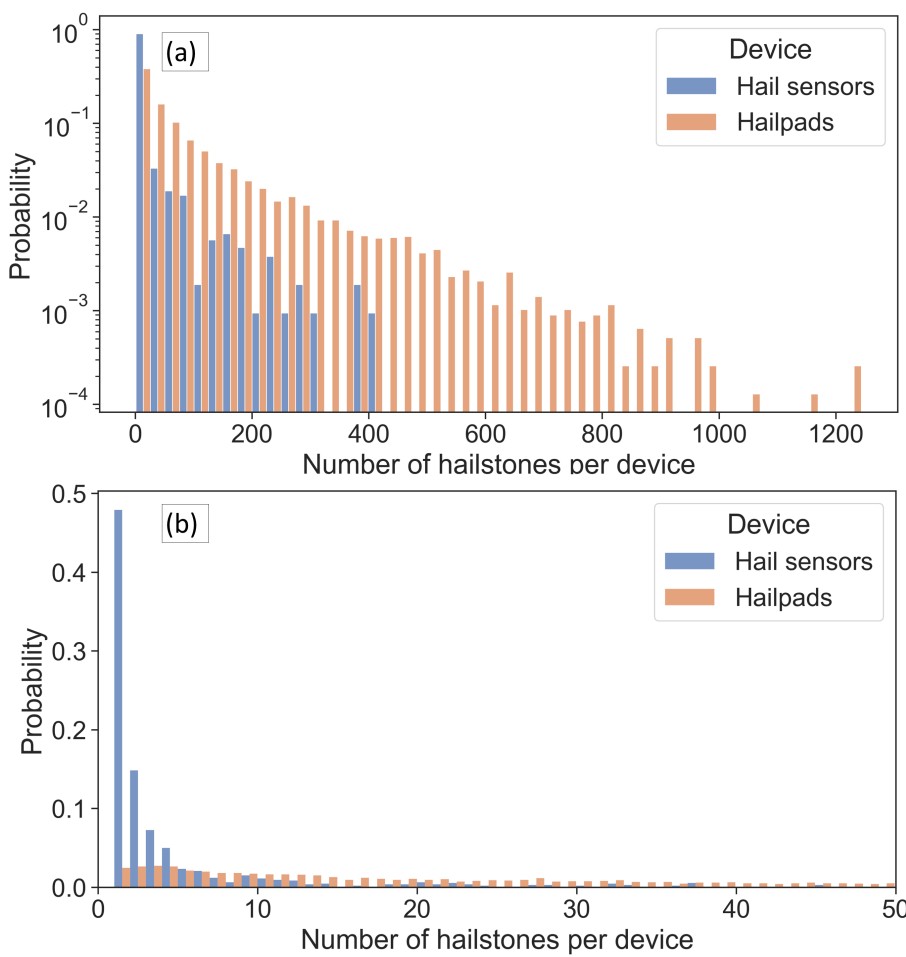

**Figure 6.** (a) Probability distributions (logarithmic y-scale) of the number of hailstones per hail sensor (orange) and hailpad (blue). (b) Same distributions as on the top with a linear y-scale and a focus on hailstones numbers from 1 to 50.



### 3.2.2 Hailstone size distributions

The hailstone size distributions (HSD) observed at the ground by the hail sensors and the hailpads are compared in Fig. 7. A visual inspection of the curves reveal that they are very similar for diameters up to 18 mm, with a slight difference for the smallest diameters. The distributions start to differ for diameters larger than 18mm (Fig. 7a). However, there is only a limited

number of data points in the hail sensor sample for those diameters, explaining the discontinuities of the distribution. Using a fit of the form $a \cdot 10^{-bD}$, similar to Eq. 7 of Manzato et al. (2022), we found:

$$P(D) = 4.84 \cdot 10^{-0.22D} \quad \text{for hail sensors} \tag{4}$$

$$P(D) = 4.44 \cdot 10^{-0.21D} \quad \text{for hailpads}, \tag{5}$$

which work reasonably well for diameters up to 18 mm but underestimate the largest hailstone probabilities.

We looked at various statistical tests to assess the similarity of the hail size distributions from hail sensors and hailpads. Figure 8 shows a quantile-quantile plot (Q-Q plot) of the two distributions. Each dot represents a percentile and we see that all dots except the last one are on, or very close to the red line, showing the similarity of the distributions up to the 99th percentile. The difference in the maximum diameter of the distributions (33 mm for hail sensors, 46 mm for hailpads) explains the large difference in the 100th percentile. A Kolmogorov-Smirnov test (statistic = 0.03, p-value = 1) did not reject the null hypothesis

that distributions are similar. A Standardized Mean Difference of 4.6672e-03 showed that the means of the distributions are comparable. A more stringent Chi-Squared test on all percentiles of the distributions rejected the null hypothesis that the distributions are the same (statistic=149.95, p-value=0.0007). However, the null hypothesis was not rejected at a 5% confidence level when considering the first 95 percentiles of the distributions in the Chi-Squared test (statistic=113.44, p-value=0.0840). Based on those tests, we conclude that the hail size distributions (HSDs) are very similar except for the tail.

The similarity of the HSDs is particularly interesting when considering the observations of the previous section. First, it shows that despite the fact that the distributions of the number of hailstones per hail sensors and hailpads are different, their overall hailstones size distribution are almost identical. Second, it shows that the two devices give coherent results when several events are pooled. We cannot draw any further conclusion, as hail sensors and hailpads observations were not made on the same hailstorms. Moreover, they were made in different regions and different years. It would also be interesting to redo this

comparison when the sample size from hail sensors becomes larger, to see if the difference in the tails remains.

   We remark here that due to their limited area, hailpads and hail sensors cannot capture the entire hail size distribution of a hailstreak. Aerial drone photography can offer promising perspectives in that sense (see for example Soderholm et al., 2020; Lainer et al., 2023), provided that one is lucky enough to be ready with a drone in the right place at the right time.



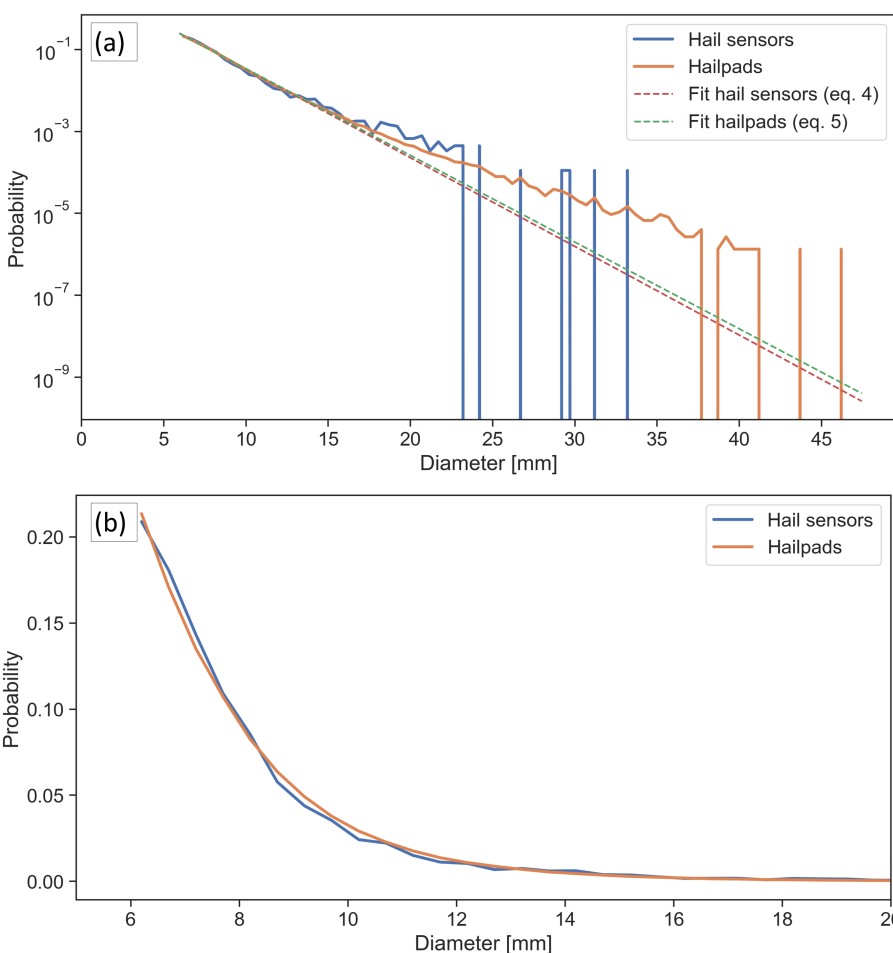

**Figure 7.** (a) Hailstones size distribution (probability) from hail sensors (orange) and hailpads (blue). The data is binned from 6 to 47 mm, using bins of 0.5 mm size (i.e. the first bin groups diameters from 6 to 6.5 mm). (b) Same distributions as in the top panel but with a linear y-scale and a focus on diameters from 6 to 20 mm.





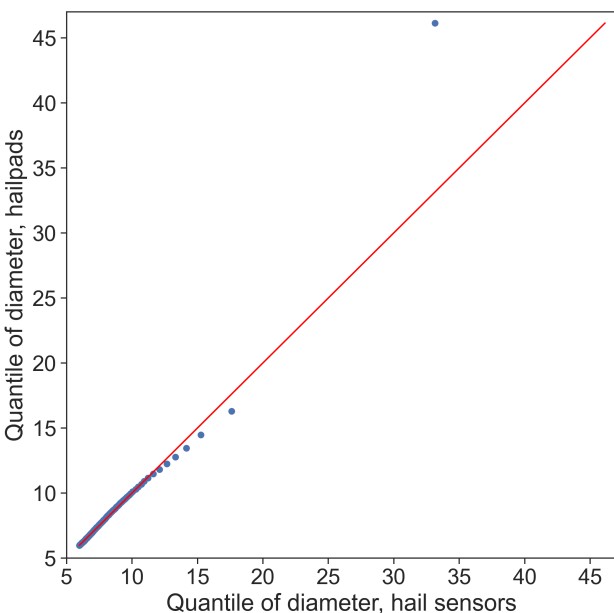

**Figure 8.** Quantile-quantile plot of the hail size distributions for hail sensors (x axis) and hailpads (y axis). Each 1% is shown.





### 3.3 Analysis of time-related quantities

We use the time information provided by the sensor to investigate the local hailfall duration, the time distribution of hailstones during an event, the hit rate (number of hailstones per second), and the relative time when the largest hailstone of an event is measured.

#### 3.3.1 Hail event duration and sensitivity analysis

We investigate hail event duration and its sensitivity to the parameter $t_{mb}$ used to delineate the events ($t_{mb}$ values of 5, 10,
15 and 20 minutes). Furthermore, we stratify the hail events by the number of impacts in 3 categories: 2 to 5, 6 to 25 and > 25 impacts. One reason is that events with very few impacts can artificially decrease the average duration as they are usually shorter than events with a larger number of impacts. Another reason is that we split events between scarce (2 to 5 impacts) and dense (> 25 impacts) hail. The value of 25 impacts to define dense events was chosen to have a sample large enough (around 100 events). We introduced an intermediate category (6 to 25 impacts) to clearly separate scarce from dense hail events. Single
impact events are not considered as their duration cannot be properly defined. From the initial 12'300 impacts, approx. 1'000 events with a single impact have been removed from the sample.

We see that the number of events and impacts in each category (first two columns of table 1) almost do not change when varying $t_{mb}$, which means that the size of the samples remain constant and are comparable.

Increasing $t_{mb}$ leads to a shift of the event duration distribution towards higher values (see Fig. 9 and table 1, columns with
event duration statistics). Considering all events (rows labelled "Total" in table 1), 50% of the events last less than 2.3 min for $t_{mb} = 5$ min and less than 3.3 min for $t_{mb} = 20$ min, corresponding to a 40% increase in median duration. 75% of the events last less than 4.4 min for $t_{mb} = 5$ min and less than 7.7 min for $t_{mb} = 20$ min, corresponding to a 75% increase in third quartile duration. Considering now only dense hail events (> 25 impacts), 50% of dense hail events last less than 3.6 min for $t_{mb} = 5$ min and less than 5.0 min for $t_{mb} = 20$ min, representing a 40% increase in median duration. 75% of dense hail events last less
than 6.0 min for $t_{mb} = 5$ min and less than 8.6 min for $t_{mb} = 20$ min, representing a 44% increase in third quartile duration. Dense hail events last longer than scarce hail events. However, quite interestingly, events in the intermediate category (6 to 25 impacts) can last longer than dense hail events, the third quartile values (75%) being always higher for the intermediate category.

We now compare our results with the existing literature. Most estimates of hailfall duration were made by human observers,
with the exception of (Changnon, 1970) and (Federer and Waldvogel, 1975). Changnon (1970) found an average duration of 3.1 min for 786 hailfalls recorded in Central Illinois (USA) from 1967 to 1968 by a raingage-hailpad network. Our average durations considering all events (table 1, 5th column) range from 3.2 min for $t_{mb} = 5$ min to 6.5 min for $t_{mb} = 20$ min. Those average durations increase to 4.8 min for $t_{mb} = 5$ min and 7.4 min for $t_{mb} = 20$ min when considering only dense events. The short durations observed by Changnon (1970) compared to ours could be explained by the relatively high average storm speeds
that they recorded: around 50 km/h, which is significantly higher that the values observed in Switzerland (see section 2.2.1).



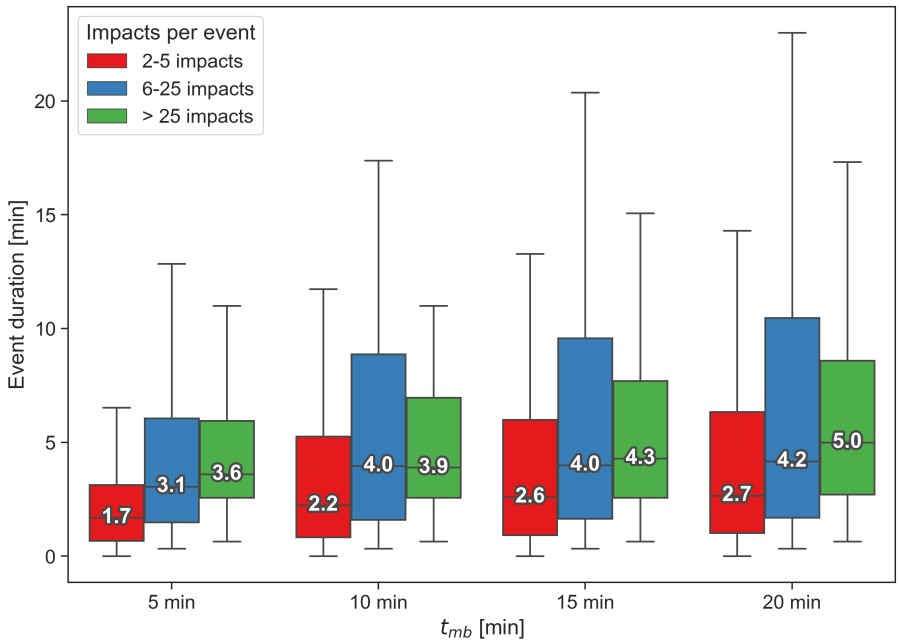

**Figure 9.** Boxplots of event duration for $t_{mb}$ values of 5, 10, 15 and 20 min, each stratified by impacts numbers (colors). Number shows the median duration in minutes.

Počakal et al. (2009) found an average duration of 4.1 min from 11500 reports on the occurrence of hail collected at hail suppression stations in Croatia between 1981-2008 by human observers. This value falls within the range of our average values. Počakal et al. (2009) mentioned that local orography can cause a decrease of storm speed thereby potentially increasing local hailfall duration. Nađ et al. (2021) discuss hail events observed in Serbia between 1981 to 2015 by human observers at hail

suppression stations. They found that hailfall lasted less than 5 min in about 75% of events, which is lower than our estimates, except in the case of scarce hail events for $t_{mb} = 5$ min (3.1 min). They also found that only 8% of events lasted more than 10 min, while in our case it ranges from 3.3% (for $t_{mb} = 5$ min) to 18.6% )for $t_{mb} = 20$ min). To our knowledge, the study of Federer and Waldvogel (1975) is the only one to have explicitly measured local hailfall duration in Switzerland for a single hailstorm on July 6, 1973 in the Napf region. Using a hailstones spectrometer, they measured a duration of 13.5 min for an

average storm speed of 6 km/h, which is relatively slow.

Our estimates of local hailfall duration in Switzerland are generally longer than durations observed in other countries by previous studies. The storm velocity, which is influenced by orography, seems to be a key factor to be further investigated. The fact that an automatic hail sensor records every single hailstone impact is another important factor. The event duration is sensitive to isolated impacts recorded by the sensors that can be missed or deemed as not being part of the bulk of a hail event

by human observers (see section 3.2.1). For this reason, we looked at another relevant quantity: the cumulative time distribution of impacts during an event.



Atmospheric
Measurement
Techniques



Discussions

**Table 1.** Event statistics for different ($t_{mb}$) values stratified by impact numbers. From left to right: number of events and hailstone impacts; mean value, first quartile, median value, third quartile and maximum value of event duration in seconds; first quartile, median value and third quartile of the cumulative time distribution of impacts (CTDI) in seconds.

| $t_{mb}$ [min] | Event impacts range | Events | Impacts | Event duration [min] | | | | | CTDI % [min] | | |
|---|---|---|---|---|---|---|---|---|---|---|---|
| | | | | mean | 25% | 50% | 75% | max | 25% | 50% | 75% |
| 5 min | 2 to 5 impacts | 330 | 873 | 2.2 | 0.7 | 1.7 | 3.1 | 9.6 | 0.0 | 0.3 | 1.8 |
| | 6 to 25 impacts | 122 | 1449 | 4.5 | 1.5 | 3.1 | 6.1 | 23.9 | 0.4 | 1.1 | 3.0 |
| | > 25 impacts | 103 | 8818 | 4.8 | 2.6 | 3.6 | 6.0 | 23.6 | 1.1 | 1.9 | 3.4 |
| | Total | 555 | 11140 | 3.2 | 1.1 | 2.3 | 4.4 | 23.9 | 0.8 | 1.7 | 3.3 |
| 10 min | 2 to 5 impacts | 345 | 932 | 3.5 | 0.8 | 2.2 | 5.3 | 15.6 | 0.0 | 0.5 | 2.8 |
| | 6 to 25 impacts | 123 | 1477 | 6.5 | 1.6 | 4.0 | 8.9 | 33.3 | 0.5 | 1.3 | 4.1 |
| | > 25 impacts | 104 | 8865 | 5.5 | 2.6 | 3.9 | 7.0 | 23.6 | 1.1 | 2.0 | 3.9 |
| | Total | 572 | 11274 | 4.5 | 1.4 | 3.0 | 6.3 | 33.3 | 0.9 | 1.9 | 3.9 |
| 15 min | 2 to 5 impacts | 342 | 934 | 4.3 | 0.9 | 2.6 | 6.0 | 30.0 | 0.0 | 0.6 | 3.1 |
| | 6 to 25 impacts | 124 | 1502 | 8.6 | 1.7 | 4.0 | 9.6 | 55.7 | 0.5 | 1.3 | 4.7 |
| | > 25 impacts | 103 | 8882 | 6.2 | 2.6 | 4.3 | 7.7 | 23.6 | 1.1 | 2.1 | 4.4 |
| | Total | 569 | 11318 | 5.6 | 1.4 | 3.1 | 7.1 | 55.7 | 0.9 | 1.9 | 4.3 |
| 20 min | 2 to 5 impacts | 339 | 925 | 4.7 | 1.0 | 2.7 | 6.3 | 34.7 | 0.0 | 0.7 | 3.3 |
| | 6 to 25 impacts | 126 | 1531 | 10.6 | 1.7 | 4.2 | 10.5 | 114.4 | 0.5 | 1.4 | 5.3 |
| | > 25 impacts | 99 | 8887 | 7.4 | 2.7 | 5.0 | 8.6 | 35.4 | 1.2 | 2.2 | 4.8 |
| | Total | 564 | 11343 | 6.5 | 1.4 | 3.3 | 7.7 | 114.4 | 0.9 | 2.0 | 4.7 |



### 3.3.2 Cumulative time distribution of hailstones impacts

The cumulative time distribution of impacts (CTDI) during an event indicates the proportion of impacts recorded before a certain duration. It is similar to a cumulative distribution function in probabilities. The last three columns of table 1) contain
the first quartile, median and third quartile values of the consolidated CTDI, while Fig. 10 shows the entire CTDI, for the four values of $(t_{mb})$, stratified by impact number categories.

We see that the third quartile of the CTDI, when 75% of impacts have been recorded (black line in Fig. 10), is reached between 3.3 min (for $t_{mb}$ = 5 min), and 4.7 min (for $t_{mb}$ = 20 min). In the previous section, we found that 75% of events last less than 4.4 min for $t_{mb}$ = 5 min and less than 7.7 min for $t_{mb}$ = 20 min. This means that the majority of impacts occurs
in a shorter time than the corresponding fraction of events, and that the majority of impacts is on average concentrated at the beginning of an event.

This is interesting because hail sensors record hailstreaks at all stages of their lifecycle (initiation, maturity, dissipation) and at all relative positions (border or center of the streak). Considering several hailstreak records, we expected the CTDI to increase steadily at an approximately constant rate, reflecting the average of the different lifecycle stages and relative positions
of all hailstreaks. However, our results show that the CTDI increases more rapidly in the beginning of an event than towards the end. Another way of saying it, is that the hailstone density (hailstone per $m^2$) is on average higher in the beginning of an event than towards the end. As an hypothesis to explain this pattern, we suggest that hailstreaks are composed of a short and intense maturing phase, where most hailstones are reaching the ground in very close succession, followed by a longer dissipating phase with a few remaining hailstones of much lower density (dissipating phase). In such a model, the odds of recording an event
with an extended phase of low hailstone density in the beginning and a phase of high density towards the end would be very low. This interpretation of the observations should be further investigated, considering our limited sample size.

It is also interesting to notice that the CTDI reaches 75% in a shorter time for dense hail events (> 25 impacts: Fig. 10, green curves) than for events in the intermediate category (6 to 25 impacts: Fig. 10, blue curves), for all $t_{mb}$ values except 5 min. The steeper CTDI for dense hail events suggests that their number of impacts per unit of time (or hit rate) is larger. This is
coherent with the findings of section 3.3.1 that events with 6 to 25 impacts can last longer than events with > 25 impacts. This is confirmed by the scatterplot of Fig. 11, showing each event for $t_{mb}$ = 10 minutes, by average hit rate (x axis), number of impacts (y axis) and the maximum instantaneous hit rate (color and size). The average hit rate is the average number of impacts per second recorded during an event, whereas the maximum instantaneous hit rate is computed as the inverse of the shortest time between two impacts recorded during the event. We use this maximum hit rate to estimate the peak intensity of the hailfall.
According to Fig. 11, events with the largest number of impacts have also the largest peak intensity. The maximum value of 15 hits per second, corresponding to the minimum dead time (see section 2.1.5), is reached only by dense events with more than 20 impacts. Events with a few impacts (scarce hail) may show a large average hit rate because on average their duration is shorter than events with more impacts, but their peak intensity is lower than those of dense hail events.





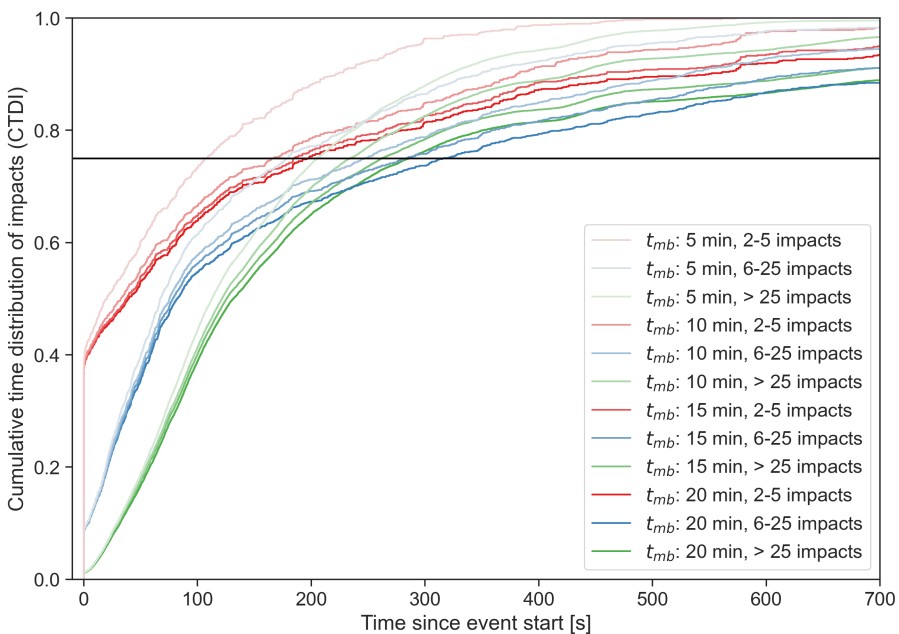

**Figure 10.** Cumulative time distribution of impacts for $t_{mb}$ ranging from 5 min (light color) to 20 min (dark color), and stratified by event hits range (colored curves). The horizontal black line denotes 75% of impacts.

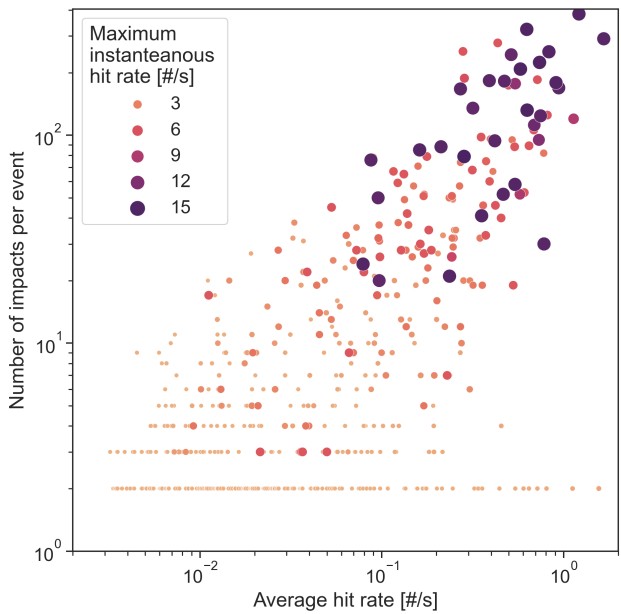

**Figure 11.** Scatterplot of hail events for $t_{mb} = 10$ min, by average hit rate (x axis) and number of impacts (y axis). Color and size represents the maximum instantaneous hit rate of the event (larger values are represented by darker and larger dots).





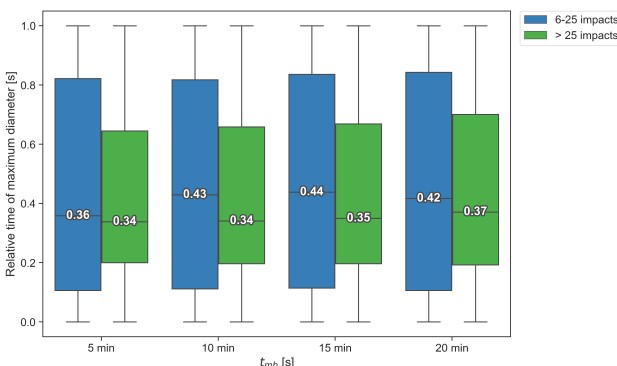

**Figure 12.** Boxplots of the relative time of occurrence of the largest hailstone during an event for $t_{mb}$ values of 5, 10, 15 and 20 min, each stratified by impacts numbers (colors). The numbers show the median relative time in seconds.

### 3.3.3 Timing of the largest hailstone ($D_{max}$).

Fig. 12 shows the distribution of the relative time when the largest hailstone ($D_{max}$) of the event hit the sensor for the four values of ($t_{mb}$). For a given event, a value of 0.5 would mean that the $D_{max}$ hailstone hit the sensor in the middle of the event.

If the $D_{max}$ hailstone hit the sensor randomly during an event then the distribution of Fig. 12 would be uniform with a 0.5 median relative time. However, as the number of impacts per event increases, the time at which $D_{max}$ occurs gets closer to the event start. For dense hail events (> 25 impacts), $D_{max}$ occurs within the first third of the event for 50% of the events (0.34

median relative time). This means that it is more likely for $D_{max}$ to be recorded in the beginning of the event. This could be explained by the fact that the larger the hailstone the faster it falls and therefore larger hailstones should hit the ground before smaller ones.

However, and as stated in the previous section 3.3.2, a hailstreak can be recorded at any stage of its lifecycle and any relative position. If the hailstones produced during a streak were first small then growing in size to reach a maximum diameter (phase of increasing diameter) and then becoming smaller (phase of decreasing diameter), and if both phases lasted equally long, we

would expect that the relative time of $D_{max}$ would be equally distributed with a mean of 0.5 for a large number of observed events. Indeed, in the phase of increasing diameters, the sensor would measure $D_{max}$ in the middle or towards the end of the event. In the phase of decreasing diameter, the sensor would measure $D_{max}$ in the beginning. Our results suggest that it is more likely for $D_{max}$ to be recorded in the beginning of the event.

This is consistent with the hypothesis formulated in section 3.3.2, that most hailstones are reaching the ground in a short and intense maturing phase, followed by a few remaining hailstones during a longer dissipating phase. In such a model, the odds of observing any hailstone (including the one with the largest diameters) in the beginning of the event is increased.

We also note that for hail events with few impacts (2 to 5 impacts) it is difficult to reach any conclusion as the number of hailstones observed is very small for each event (not shown).



## 4 Summary, conclusions and outlook

We present an analysis of automatic hail sensors data from a national network in Switzerland. Our study is based on a sample of about 12'000 hailstone impacts and 500 hail events, gathered during 3 to 5 hail seasons, depending on the sensor location. The capacity of the sensors to record the precise timing of hailstone impacts and their kinetic energy opens new research avenues on local hailstorms duration and lifecycle, and time-resolved hailstone size distributions.

As any measurement device, the sensors come with some limitations: an uncertainty on the diameter measurement for which it's recommended to work with 5mm size bins; a diameter dependent dead time ranging from 0.064 s to 0.5 s which can result in 4% to 4.6% of missed impacts; and the possible recording of impacts not due to hail. This accuracy is sufficient for our study.

The impacts of those limitations could be reduced by using the hail sensors together with other hail measurement devices. The number and size of the missed impacts could be estimated by using hailpads in conjunction with hail sensors and comparing their size distribution. This could also help reducing the uncertainty in the diameter measurements. The use of advanced radar-based hail algorithms (such as ZDR columns or hydrometeor classification, see e.g. Besic et al. (2016)) could help discriminating impacts not due to hail.

More specifically, we recommend performing the field calibration and resetting the minimum threshold right before installing or using the sensor in a field campaign to get the most precise measurements. If two sensors are deployed at the same location, it could also be interesting to set a signal threshold corresponding to diameters lower than 5 mm on one of them to observe graupel.

We showed that despite their different measurement approaches, hail sensors and hailpads measure the same hail size distributions, as confirmed by the similar fits of Eq. 5 and Eq. 4 and several statistical tests.

We discuss time series of impacts recorded by the hail sensors to illustrate that the definition of a hail event is sometimes not trivial, and propose a method using a single parameter $t_{mb}$ to characterize the events. Recommended values for $t_{mb}$ are between 5 min and 20 min.

The timing of hailstone impacts can be used to extract the local duration of hailfalls. The majority of local hailfalls last just a few minutes (less than 4.4 min for $t_{mb} = 5$ min and less than 7.7 min for $t_{mb} = 20$ min). Those durations are slightly higher than previously reported durations by human observers in the literature for other countries. The difference might be due to the high sensitivity of the hail sensor, which records every single impact, compared to a human observer, who can miss those single impacts. To focus on the bulk of hail events, we looked at the cumulative time distribution of impacts (CTDI) and find that the majority of impacts occurred in less than 3.3 min ($t_{mb} = 5$ min) to less than 4.7 min ($t_{mb} = 20$ min), a range of duration that is comparable to existing literature. The fact that local hailfall duration is usually shorter than 5 minutes, should also be considered when examining radar based data, as their time resolution in Switzerland is 5 minutes.

The majority of the hailstone impacts is on average concentrated at the beginning of an event. This suggests that most events are composed of a rapidly increasing and short phase of high hailstone density at the beginning of the event followed by a longer phase with low hailstone density. Most hailstones, including the largest, fall during the first phase of high intensity,



while a few remaining and smaller hailstones fall in the second low density phase. This interpretation of the observations
remains an hypothesis, and should be further investigated.

Finally, we observed that the hail events with the largest number of impacts have also the largest peak intensity as measured by the maximum instantaneous hit rate.

The sensor data provide first interesting results but more observations are needed to reach more robust conclusions. The sensor network in Switzerland will be operating for at least 4 more years and other countries have started or consider using the
same hail sensor model for national networks.

Our comparison of hail sensor and hailpad observations is done using samples coming from two different regions composed of a different set of hailstorms. Ideally hail sensors and hailpads should be paired at the same locations for comparison. Used in pairs they would not only give a more complete view of the hail size distribution, but would also allow for the identification of possible differences in measurements between the two devices and thus get a better understanding of their respective biases.

Hail is a rare phenomena, extremely localised in both space and time, and therefore challenging to observe. Automatic hail sensors allow to observe hail extremely precisely in time, something which was difficult if not impossible with existing devices. However, they should be used together with other measurement sources such as hailpads, (mobile) radar, crowdsourced observations or drone aerial measurements. Such a combination would allow to get the most comprehensive picture of a hailstreak and its related hailstones distribution in further research.



*Code availability.*   The python code used in this study is available on the following github page: https://github.com/jekopp-git/hailsensors_ observations

*Author contributions.*   JK: conceptualization; methodology; hail sensor data preparation and validation; code; statistical and formal analysis; visualisations; writing original draft. AM: conceptualization; hailpads data preparation and validation; review and editing. AH, UG, OM: conceptualization, review and editing.

*Competing interests.*   The authors declare that they have no conflict of interest.

*Acknowledgements.*   We thank the Swiss Insurance Company La Mobilière for funding the automatic hail sensors network and making the hail sensor data available for these analyses. We thank Daniel Wolfensberger (MeteoSwiss) for providing the sensor data. We thank Serge Mattli (inNET AG) for helpful discussions on the technical aspects of the hail sensors. Jérôme Kopp and Olivia Martius acknowledge support from La Mobilière. Olivia Martius acknowledges support from the Swiss Science Foundation (grant no. 201792).



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
