# Peer review of "How observations from automatic hail sensors in Switzerland shed light on local hailfall duration and compare with hailpad measurements"

_Atmospheric Measurement Techniques, 2023_

## Referee Comment (RC1)

Review of "How observations from automatic hail sensors in Switzerland shed light on local hailfall duration and compare with hailpads measurements", by Kopp, Manzato, Hering, Germann and Martius, AMT-2023-68.

This is a worthwhile study that reports on data from automatic hail sensors based on the Loffler-Mang et al. (2011) instrument. There were 80 of these automatic hail sensors operational in Switzerland, some of which acquired data from 2018-2022. The observations were compared with data from a hailpad network in northern Italy. Some interesting aspects of the hail sensor data were examined, including the duration of hail, when the largest hail was observed relative to the beginning of the hail event, etc. My comments appear below. I recommend that the revision considers my comments below.

**Main Comments**

This article has many similarities to the Kopp et al. (2022) article that you cite. You do mention that in the text. Perhaps you could add a sentence on some of the results from that article.

I tried to find the Wetzel (2018) article in MTI but couldn't. You may want to add references to the instrument that appeared in the Kopp et al. (2022) article.

Eq. (1). A square root sign when a variable is taken to the ¼ power. Please convert it to $[\quad]^{0.25}$

When a comma, should be used, for example 10,200 you have 10'200. There are many places where this needs correction. For example, line 43: 12'300

Using the values you provided for the variables in Eq. (1), I did a curve fit that relates diameter to kinetic energy. The goodness of fit is almost 1.00. I did the same thing using the values presented in Heymsfield et al. (2018). As you can see from the figure below, the spherical assumption results in hailstone diameters that are perhaps 30% *smaller*. It took me a while to figure out why but after thinking about it is definitely the case. In Figure 12 of the Kopp et al. (2022) article, there are good examples of why you should consider non-spherical particles. I feel extremely strongly that we need to move forward acknowledging that the assumption of

spherical hailstones needs to be replaced with current knowledge-not citing studies from the 1970's. I suggest that you include both spherical and the "nonspherical" hailstone assumptions in your discussion (lines 98-100, etc.)

Section 2.1.6. What if the hailstones 5 to 7 mm are non-spherical. They could be considered raindrops?

Could you add nonspherical hailstone diameters to Fig. 6a. Would this improve the agreement between the hail sensors and hailpads?

As you know, probability is not concentration. The concentration is the number per cubic volume (meters). Could you estimate the terminal velocity to get the concentration? That would be useful information. In fact, in this way you could compare your hail size distributions to the article sby Federer and Waldvogel, and others from the 1970s. Also, the impacts on the hailpads are not likely to be spherical because of the non-spherical shape of large hailstones. In fact, the shape of the impacts could be examined for non-sphericity. This has been done in the past.

I like the idea of characterizing the duration of the hailstorm events at the ground and its distribution with time. Could you possibly link radar data to your hail impact data and then in the future be able to use radar data to refine the estimates of hail duration?

3.3.3 Timing of the largest hailstone. Just a thought. Prior to the largest hailstone, what do you observe in the rain category? Perhaps the smaller hailstones melt prior to reaching the surface and only the larger ones survive the melting process.

**Minor Comments**

Title. Hailpads should be singular, Hailpad.

Lines 1, 15. Measuring the properties of hailstorms

28. "cheap" to "low cost"

160-161. I very much like the idea of using radar to identify when hailstorms are in close proximity to your sensors.

175. Canada or the United States? I didn't have access to the Brimelow study

205. Good idea to look at a range of time intervals

Figure 5a. Put in the alternate diameter. This would be very useful here.

437-438. Could there be video cameras that are turned on and off by the hail sensors when they detect hail? That might provide another means of characterizing the hail events.

More of my comments will be given in a revision.

Andy Heymsfield, NCAR

[Figure]

Fig. 1

---

## Author Comment (AC1)

Review of "How observations from automatic hail sensors in Switzerland shed light on local hailfall duration and compare with hailpads measurements", by Kopp, Manzato, Hering, Germann and Mar/us, AMT-2023-68.

This is a worthwhile study that reports on data from automatic hail sensors based on the Loffler- Mang et al. (2011) instrument. There were 80 of these automatic hail sensors operational in Switzerland, some of which acquired data from 2018-2022. The observations were compared with data from a hailpad network in northern Italy. Some interesting aspects of the hail sensor data were examined, including the duration of hail, when the largest hail was observed relative to the beginning of the hail event, etc. My comments appear below. I recommend that the revision considers my comments below.

We thank the referee for its revision and both supporting and pertinent comments.

**Main Comments**

This article has many similarities to the Kopp et al. (2022) article that you cite. You do mention that in the text. Perhaps you could add a sentence on some of the results from that article.

Kopp et al. (2022, Weather) describe the weather situations that let to severe hailstorms in summer 2022 and contains a qualitative description of the hail sensor measurements and comparison to crowd source data gathered on 8. July 2021. The present article presents a detailed discussion of the entire hail sensor data set and a detailed comparison with hail pad measurement.  Hence the research questions of Kopp et al. (2022) differ from those of the present article. We agree with the referee that we should include some of the conclusions of Kopp et al. (2022) regarding automatic hail sensors:

Line 44: Some observations recorded during the particularly active hail season of 2021 were presented in Kopp et al. (2022) **where it was shown that automatic hail sensors could successfully capture precise time series of individual hailstone impacts.**

Line 230: The highest number of yearly impacts **(6'400)** was recorded in summer 2021, during a particularly active hail season (Kopp et al., 2022).

I tried to find the Wetzel (2018) article in MTI but couldn't. You may want to add references to the instrument that appeared in the Kopp et al. (2022) article.

We agree with the referee that the access of the reference is not straightforward. The references of Kopp et al. 2022 related to the hail sensor are the same as those of the present article.  The reference Wetzel, 2018 can be found here: https://www.innetag.ch/wp-content/uploads/2020/10/HailSens-MTI_2018_09.pdf

We made the following change:

Url added to the reference in the bibliography

Eq. (1). A square root sign when a variable is taken to the ¼ power. Please convert it to [ ]$^{0.25}$

We modified accordingly with []$^{0.25}$.

When a comma, should be used, for example 10,200 you have 10'200. There are many places where this needs correction. For example, line 43: 12'300

We agree with the referee that this notation is not an official one.

Change: We replaced the symbol ' by a space, following guidelines from the International System of Units (SI).

Using the values you provided for the variables in Eq. (1), I did a curve fit that relates diameter to kinetic energy. The goodness of fit is almost 1.00. I did the same thing using the values presented in Heymsfield et al. (2018). As you can see from the figure below, the spherical assumption results in hailstone diameters that are perhaps 30% *smaller*. It took me a while to figure out why but after thinking about it is definitely the case. In Figure 12 of the Kopp et al. 2022) article, there are good examples of why you should consider non-spherical particles. I feel extremely strongly that we need to move forward acknowledging that the assumption of spherical hailstones needs to be replaced with current knowledge-not citing studies from the 1970's. I suggest that you include both spherical and the "nonspherical" hailstone assumptions in your discussion (lines 98-100, etc.)

We agree with the referee that considering hailstones to be spherical is an approximation, which we explicitly acknowledge on lines 91 to 96. We used the output diameters of the instrument as provided by the manufacturer (ie. Using the spherical approximation), and changing this internal estimation is beyond the scope of this work. Moreover, applying non-spherical approaches would require further steps to be properly conducted. First, it would require information on the aspect ratios or sphericity of the measured hailstones, which unfortunately is not given by the hail sensors. Using hailpad and hail sensor in pairs could help fill this gap. Second, both automatic hail sensors and hailpads are calibrated using spherical masses. The calibration process would need to be conducted using masses of different shapes to properly assess the impact of the non-sphericity on the output of both devices (electric signal and dents).

Besides, we note that the hailstones shown on Figure 12 of Kopp et al. (2022) have (largest) diameters between 4cm and 10cm, which are larger than the largest hailstone measured by the automatic hail sensors (3.3cm).

We made the following changes in our conclusion:

Line 100: **We discuss this point further in the conclusion.**

Line 435 (new paragraph). **We used the output diameters of the hail sensor as provided by the manufacturer in the present study. Such diameters were estimated using the approximation that hailstones are spherical, which is not always the case, especially for large hailstones. Pairs of hailpad and hail sensor could be used to investigate more**

**advanced non-spherical approaches \citep[eg.][]{Heymsfield2018, Shedd2021}. For example, the distribution of aspect ratios could be inferred from the hailpad measurements and used as an input when estimating the hailstone dimensions from the kinetic energy measurements of the hail sensors.**

Section 2.1.6. What if the hailstones 5 to 7 mm are non-spherical. They could be considered raindrops?

We agree with the referee that there is an overlap between the range of kinetic energies of small hailstones and very large rain drops. Consequently, we cannot exclude that some of the recorded impacts were caused by large rain drops and not by hailstones. However, we believe this effect to be limited because the non-spherical hailstones should be the largest and not those of 5-7mm. Then, if we consider all (or a large fraction of) hailstones from 5 to 7 mm as raindrops, then our hail size distribution would become very different from the size distribution measured by hailpads (for small sizes) as hailpads do not record raindrops. While the two distributions may differ due to several factors, we think as rather unlikely that the number of hailstones between 6mm and 7mm is much lower in Switzerland than in Italy.

Could you add nonspherical hailstone diameters to Fig. 6a. Would this improve the agreement between the hail sensors and hailpads?

As mentioned in our answer above, we believe that using non-spherical assumptions is a valuable research avenue but that it requires a more detailed analysis which is beyond the scope of the present paper. We mention in our conclusion how this could be further conducted.

As you know, probability is not concentration. The concentration is the number per cubic volume (meters). Could you estimate the terminal velocity to get the concentration? That would be useful information. In fact, in this way you could compare your hail size distributions to the article by Federer and Waldvogel, and others from the 1970s.

We agree with the referee that we limited our study to the hail size distribution measured at the ground, because our objective was the comparison of the measurements of hailpads and hail sensors and for that the hailstone concentration was not needed. We are not sure which publication of Federer and Waldvogel the referee is mentioning. If such references analyse the hailstone concentration within the cloud, then assumptions on the melting rate and hailstone drifting would have to be made. This is another interesting research question which could be treated in another publication but is beyond the initial scope of the present paper.

Also, the impacts on the hailpads are not likely to be spherical because of the non-spherical shape of large hailstones. In fact, the shape of the impacts could be examined for non-sphericity. This has been done in the past.

A discussion on the aspect ratios of the impacts on the hailpads is available in Manzato et al., 2022. They found that the median aspect ratio for hailstones larger than 6mm is approx. 1.25. The distribution of the aspect ratios is available in their figure 3.

Manzato, A., Cicogna, A., Centore, M., Battistutta, P., and Trevisan, M.: Hailstone characteristics in NE Italy from 29 years of hailpad data, Journal of Applied Meteorology and Climatology, https://doi.org/10.1175/JAMC-D-21-0251.1, 2022.

I like the idea of characterizing the duration of the hailstorm events at the ground and its distribution with time. Could you possibly link radar data to your hail impact data and then in the future be able to use radar data to refine the estimates of hail duration?

We thank the referee for this suggestion. Using radar data to refine the estimates of the hail duration would not be feasible due to the current spatial (1 km$^2$) and temporal (5 minutes) of the Switzerland radar network. We mention at line 419-420 that the local duration of most hailfalls is less than this temporal resolution.

3.3.3 Timing of the largest hailstone. Just a thought. Prior to the largest hailstone, what do you observe in the rain category? Perhaps the smaller hailstones melt prior to reaching the surface and only the larger ones survive the melting process.

It would be interesting to analyse this point in more details. Melting certainly affected the hailstone sizes during the events. However, no rain disdrometer was installed next to the hail sensors to measure the rain drop size distribution. Moreover, the time (5 min) and space (1km) resolution of radar-based estimates of the rain rate would also be too coarse to make a proper estimation at the sensor location. Therefore, we are not able to answer this question.

**Minor Comments**
Title. Hailpads should be singular, Hailpad.

We changed accordingly.

Lines 1, 15. Measuring the properties of hailstorms

We changed accordingly.

28. "cheap" to "low cost"

We replaced cheap with affordable.

160-161. I very much like the idea of using radar to identify when hailstorms are in close proximity to your sensors.

175. Canada or the United States? I didn't have access to the Brimelow study

We thank the referee for pointing this out. Brimelow (2018) refers to the following studies which analysed hailstorms in various countries:

Admirat, P., Goyer, G. G., Wojtiw, L., Carte, E. A., Roos, D., & Lozowski, E. P. (1985). A comparative study of hail in **Switzerland, Canada and South Africa**. Journal of Climatology, 5, 35–51.

Frisby, E. M. (1963). Hailstorms of **the Upper Great Plains of the United States**. Journal of Applied Meteorology, 2, 759766.

Nelson, S. P., & Young, S. K. (1979). Characteristics of **Oklahoma** hailfalls and hailstorms. Journal of Atmospheric Sciences, 18, 339–347.

Paul, A. H. (1980). Hailstorms in southern **Saskatchewan**. Journal of applied Meteorology, 19, 305–314.

Webb, J. D. C., Elsom, D. M., & Meaden, G. T. (2009). Severe hailstorms in **Britain and Ireland**, a climatological survey and hazard assessment. Atmospheric Research, 93, 587–606.

We made the following change:

Line 175:  **Based on studies from various countries, Brimelow (2018) found** that the majority of hailstreaks are less than 5 km wide, increasing to 10 to 15 km for more organized hailstorms, with maximum widths ranging from 25 to 30 km

205. Good idea to look at a range of time intervals

Figure 5a. Put in the alternate diameter. This would be very useful here.

We refer to our comment on non-spherical hailstones above.

437-438. Could there be video cameras that are turned on and off by the hail sensors when they detect hail? That might provide another means of characterizing the hail events.

This is an interesting idea, provided that the camera is properly shielded from the hailstones. The associated cost of adding, setting up and maintaining the cameras would also need to be considered.

More of my comments will be given in a revision.
Andy Heymsfield, NCAR

[Figure]

Fig. 1

---

## Author Comment (AC2)

Overview

The authors present results from a hail disdrometer network deployed over three regions in Switzerland. Although piezoelectric sensors have been used before to estimate hail size, this paper improves our understanding of the performance of this instrument, as well as its strengths and weaknesses for monitoring hail. The paper is well written and organized. By my reckoning, it also meets the criteria for publication in AMT.

We thank the referee for his detailed revision and relevant comments.

Technical Comments

Page 2, line 25: Please add the work undertaken by Strong and Lozowski in Alberta.

We added the following reference:

Lozowski, E. P., and G. S. Strong, 1978: On the Calibration of Hailpads. *J. Appl. Meteor. Climatol.*, **17**, 521–528, https://doi.org/10.1175/1520-0450(1978)017<0521:OTCOH>2.0.CO;2.

Page 2, line 29: This discussion is too simple, even for an overview. Please discuss in more detail the factors that affect the size or dents left by hail on hailpads (e.g., physical properties of the hail, properties of the foam, etc.).

We changed the original text "Upon impact, the hailstone leaves a dent in the hailpad, whose size depends on the hailstone dimension."

With the more informative one:

**"Upon impact, the hailstone leaves a dent in the hailpad, whose size depends on the hailstone shape and density, in addition to the specific response of the hailpad material. To estimate the hailstone size from this dent it is assumed that the hailstone is spherical, has a constant density, and that the minor axe of an ellipse used to fit the dent is related to the hailstone diameter via a linear calibration fit, specific to the hailpad material (Manzato et al. 2022, Palencia et al. 2011)."**

The following reference was added:

Palencia, C., A. Castro, D. Giaiotti, F. Stel, and R. Fraile, 2011: Dent Overlap in Hailpads: Error Estimation and Measurement Correction. *J. Appl. Meteor. Climatol.*, 50, 1073–1087, https://doi.org/10.1175/2010JAMC2457.1.

Page 2, line 41: This is more than a "time-recording instrument", the text needs to reflect more accurately what it does (i.e., it measures the time and number of the impacts and an estimate of the corresponding hail).

We made the following change:

Line 42: The automatic hail sensor deployed in the network is a later version of the prototype developed by Loffler-Mang (2011). **The instrument records the precise timing of each hailstone impact and estimates their corresponding kinetic energy and diameter**.

Page 2, lines 50-51: Do you mean what new information compared to hail pads can be provided? Consider numbering the objectives.

We meant compared to existing instruments used to measure hail in general. We stated it more explicitly:

Line 50: how to make the best use of the sensor observations and what new information can they provide about hail **compared to existing instruments?**

We also changed to a numbering list.

Page 4, the sensor network. Please provide more information about the spacing of sensors in the network. What is the mean nearest neighbour distance for each network and all sub networks combined? What is the area of each sub-network and for all three networks combined?

We agree with the referee that more information could be provided. We modified accordingly:

Line 69:

The distance between neighbouring sensors varies considerably within each region. The average distance is 1.1 km for the Jura, 1.3 km for the Ticino, 3.5 km for the Napf region, and, 2.3 km for all three regions combined. The distances are short enough to have multiple sensors sampling the same hailstorm. The exact location of the sensors also depends on instrumental and practical aspects such as little shadowing and access.

Caption of Fig. 1.: The areas covered by the three networks are approximately 53 km^2 for the Jura, 440 km^2 for the Napf (excluding the three sensors in Bern, Luzern and Thun) and 86 km^2 for the Ticino.

Page 4, line 81: What are these calibrations spheres manufactured from?

They are made of polyamide. We included more details in the description of the calibration:

Line 81: **Three rods of different known masses which are screwed onto a polyamide sphere at the bottom** are each dropped twelve times from two fixed heights **on the same calibration point**. **A material factor (determined at the factory using a hail gun) considers the different impact behavior of ice and polyamide**.

Page 5, line 89: An air density of 1.2 kg m-3 seems too high for a study area that includes mountainous terrain. Would it not be more appropriate to use the observed air density for each station location and event? Alternatively, you could use the mean observed air density for a given station location.

We agree with the referee that it would be more appropriate to use observed air density values and/or to adjust the density depending on the station elevation. However, we wanted

to use the output diameters of the instrument as they are provided by the manufacturer (i.e. calculated using the listed values for the parameters). Changing the internal estimation (proprietary) made by the instrument manufacturer is beyond the scope of this work, even if it can be considered for future improvements.

Page 5, line 90: The values for the drag coefficient and bulk hailstone density are reasonable. Please provide references to support these choices.

The values for the drag coefficient and bulk hailstone density are those used by the manufacturer in the internal estimation. We made the following change:

Line 90: Diameter calculations using Eq. \ref{eq:1} **and the listed values for its parameters** are directly implemented in the hail sensor software **by the manufacturer**. **We note that values of $\rho_{air}$, $\rho_{ice}$ and $c_w$ can vary depending on the local environment and from one hailstone to another, but that similar values have been used previously in the literature (Waldvogel et al. 1978, Brimelow 2018, Manzato et al. 2022}.**

The following reference was added:

Waldvogel, A., B. Federer, W. Schmid, and J. F. Mezeix, 1978: The Kinetic Energy of Hailfalls. Part II: Radar and Hailpads. *J. Appl. Meteor. Climatol.*, 17, 1680–1693, https://doi.org/10.1175/1520-0450(1978)017<1680:TKEOHP>2.0.CO;2.

Page 6, line 98: I would argue that the effect of the drag coefficient is not limited. What are the authors basing this claim on? Please elucidate.

Here we wanted to say that we do not expect the drag coefficient to depart significantly from the 0.5 value, based on the assumption that the shape of small hailstones (<2cm) can be reasonably approximated by a sphere. We made the following change to clarify the statement:

Line 98: We focus on relatively small hailstones, most of them with an estimated diameter of less than 20 mm, such that the assumption of spherical hailstones remains a **reasonable** approximation . **Therefore, we do not expect the drag coefficient to significantly depart from the 0.5 value (Waldvogel et al., 1978; Shedd et al., 2021).**

Page 8, line 162: A threshold of 35 dBZ is very generous given that hail is typically associated with reflectivities of ~50 dBZ and higher. Is the 35 dBZ threshold value from a previous study? If so, please include the reference.

35dBZ is the common threshold used to identify environments for convective storms in Switzerland using C-band radar composites (see Section 3.5 of Nisi et al. 2016). A 35 dBZ threshold was used in Barras et al. (2019) for filtering hail crowdsourced reports. Their study showed that radar-based hail algorithms making use of a higher reflectivity threshold (45 dBZ) might be too restrictive in some cases.

We made the following change:

Line 164: … and was used in Barras et al. (2019) to filter hail crowdsourced reports. While usually hail is associated with higher maximum reflectivity in radar-based hail algorithms (e.g., 40 to 50 dBZ; Waldvogel et al., 1979, Witt 1998, Joe 2004), Barras et al. (2019) found that it might be too restrictive.

Page 8, lines 164-165: Are the number of events excluded sensitive to changing the radius to 2 km or 8 km? How many events were excluded using a radius of 4 km?

1785 hailstone impacts were excluded using a minimum reflectivity threshold of at least 35 dBZ within a radius of 4km. A radius of 10km excluded 1677 impacts and using only the reflectivity value (1 km^2 pixel area) at the sensor location excluded 2409 impacts.

We made the following change:

Line 164: This filter removed 1 785 hailstone impacts out of 14 085 from our dataset.

Page 9, event detection: Just a thought. Could one possibly objectively identify events using a temporal KDE method or a clustering technique? This would avoid having to optimize tmb.

We agree with the referee that other methods could be explored. We note that the variability in the hailstone temporal density between events and the limited number of impacts in some events could represent a challenge in the use of such methods.

Page 12, line 209: Please provide the dimensions of the hailpad associated with this surface area.

We made the following change:

Line 209: 0.291 m * 0.395 m

Page 18, comparison with hailpad data: I'm perplexed why the authors specifically selected to compare data collected in Switzerland with hailpad data from a network in Italy. It is well established in the literature that there is no unified hail-size distribution, with the results varying by geographical region (e.g., Admirat et al. 1985, J. Climate; Sanchez et al. 2009, Atmos-Res.) and even by storm environment (e.g., Cheng et al. 1985, JACM). The authors do speak to this briefly (lines 282-284), but this choice needs much more rigorous support. Further, there are hailpad data from a network of over 300 hailpads deployed in Switzerland during operation Grossversuch. Why were these data not used for the comparison instead?

We understand the referee's concerns and we agree that we should justify our choice in more detail.

On lines 210-214, we explain that it was important to have as many details as possible on the hailpad/dent selection, measurements, and processing to make the comparison with the hail sensors as close as possible.

Several papers present results and hailstones spectra derived from Grossversuch IV hailpad observations (eg. Waldvogel et al., 1978a and 1978b; Mezeix et al., 1983; Federer et al., 1986; Smith and Waldvogel, 1989; Schmid et al., 1992). However, each of these studies uses

a different subset of Grossversuch IV hailpad measurements, analyses different averaged quantities and gives only limited details on the hailpad/dents selection process. Consequently, we would have needed to have access to Grossversuch IV original data to process it and have direct contact with someone involved in (or with in-depth knowledge of) the data to get additional information to make a proper and precise comparison. Grossversuch IV (Federer et al., 1986) took place more than 40 years ago (1977-1981), and it would have been very difficult (if not impossible) to do so.

Therefore, we chose to use the data of Manzato et al. (2022) because (1) we had direct access both to the detailed data and to someone with in-depth knowledge about it, and (2) while the observations were made in regions from two different countries, they are geographically close (approx. 300km apart), at the same latitudes and both areas are close to the Alpine chain and subject to similar synoptical scale systems (Giaotti et al., 2003). Moreover, both datasets include multiple storms from several years, allowing us to average out the effects of specific storm environments.

Lastly, we found very interesting the fact that the two resulting hailstone size distributions shown in Fig. 7 are so much similar.

We made the following changes:

Line 211:

**Consequently, it makes sense to compare their observations.**

**Contrary to some of its neighboring countries, Switzerland does not have an operating network of hailpads. A network of around 300 hailpads was set up during the Grossversuch IV experiment (Federer et al., 1986) from 1977 to 1981. Several studies presented results and hailstones size distributions using Grossversuch IV hailpad observations (e.g. Waldvogel et al., 1978a and 1978b; Mezeix et al., 1983; Federer et al., 1986; Smith and Waldvogel, 1989; Schmid et al., 1992). However, each of these studies uses a different subset of Grossversuch IV hailpad measurements. They provide only averaged quantities computed from those subsets and give limited details on the hailpad selection process (e.g., is there a minimum number of dents to consider a hailpad? What is the calibration fit?), precluding a precise comparison with the hail sensor observations. The same conclusions were reached when reviewing hailpad studies covering other regions of the world (Fraile et al., 2003; Sánchez et al., 2009; Pocakal, 2011; Eccel et al., 2012).**

**For those reasons**, we chose to work with hailpad observations from a stations network of NE Italy, collected during the 1988-2016 (29 years) warm seasons (Manzato et al., 2022), **because direct access to both the detailed dataset and someone with in-depth knowledge about it was possible. While the observations were made in two different countries, Switzerland and NE Italy are geographically close (approx. 300km apart), at the same latitudes, and are close to the Alpine chain and hence subject to similar synoptical scale weather systems (Giaotti et al., 2003). Therefore, we do not expect the hail size distributions to differ substantially on average. Moreover, both datasets include multiple**

**hailstorms from several years, contributing to averaging out the effects of storm environments (Cheng et al., 1985).**

Page 18, equation (4): Can the authors please speak to the similarities and differences between equations (3) and (4). Are they essentially the same, but the form used in (4) was adopted for comparison with the results from Manzato et al. (2022)? The reason for using two equations for the same distribution needs to be made clear.

We thank the referee for pointing out the difference between equation (3) and (4). We changed the formulation of equ. (3) so that it now matches the form of equ. (4). The form of equ. (4) was indeed adopted for comparison with Manzato et al. 2022.

We made the following changes:

Equation (3) now reads:

$$P(D) = 2.48(\pm 0.42) \cdot 10^{-0.16(\pm 0.01)D}$$

As the two forms were equivalent, Fig. 5a remains unchanged.

Page 21, lines 294-296: It would probably be easier for the reader to follow the discussion if the category names were included following the impact ranges. So, "…: 2 to 5 (scarce), 6 to 25 (intermediate) and > 25 (dense) impacts.".

We agree and make the following change:

Line 294: Furthermore, we stratify the hail events by the number of impacts in 3 categories: 2 to 5 (**scarce**), 6 to 25 (**intermediate**), and > 25 impacts (**dense**).

Page 21, lines 311-313: Do the authors have any ideas as to why this might be the case?

It might be related to the hailstone density. A lower hailstone density would result in impacts more spread out over time. However, a more detailed analysis of the implied events would be necessary.

Page 22, hailfall duration: Please also reference the work of Wojtiw (1975, "Hailfall and crop damage in Alberta"). He found a mean hailfall duration of 10.1 minutes.

We thank the referee for this additional reference.

Wojtiw, L. (1975). Hailfall and crop damage in central Alberta. *The Journal of Weather Modification*, *7*(2), 28–42. https://doi.org/10.54782/jwm.v7i2.677

We made the following change.

Line 321: Wojtiw (1975) found an average duration of 10.1 min for 455 hail swaths recorded in Central Alberta (USA) from 1957 to 1973 by human observers. Their study focused on major hail swaths from which at least 10 hail reports were obtained, with large (walnut size) hail reported at some point, which might explain the longer duration.

Page 22, line 330: Not to be fussy, but I would argue that a storm speed of 6 km/h is "very slow" and not "relatively slow". A reference providing a mean storm speed would be helpful for adding context.

We made the following change:

Using a hailstones spectrometer, they measured a duration of 13.5 min for an average storm speed of 6 km/h, which is very **slow compared to the average storm speed in Switzerland (25 to 33 km/h; Nisi et al., 2018).**

Page 24, lines 350 to 356: Is this phenomenon perhaps a reflection of size sorting of hail, for example?

Size sorting certainly plays a role as shown in Fig. 12. However, if only size sorting would apply and admitting that we have more small than large hailstones in each hailstreak, then we would expect the hailfall to begin with a low-density phase composed by the largest hailstones, followed by a high-density phase made of the smallest hailstones.

Page 27, lines 411-412: Is it not possible to suggest a narrower range or a specific value for tmb? What value will the authors be using in future analyses of these data?

The largest sensitivity of the event duration and CTDI is observed when tmb increases from 5 to 10 min, whereas there is very little change when tmb increases from 15 to 20 min. Therefore, we would suggest using a 10 to 15 min value for tmb.

We made the following changes:

Line 411:

We **test** values of tmb between 5 and 20 min

Line 420:

The most considerable sensitivity of the hailfall duration and CTDI is observed when tmb increases from 5 to 10 min, whereas there is very little change when tmb increases from 15 to 20 min. Therefore, we suggest using a 10 to 15 min value for tmb in further studies.

Page 27, final section: Do the authors plan to locate hail pads or cameras next to some of the sensors in the future? This would be very helpful for evaluating the sensors.

This is something we considered. But it comes with additional costs (hailpad collection, replacement, camera installation). To complement the hail measurements of the sensors MeteoSwiss is experimenting with a drone that is equipped with a high-resolution camera.

Page 27, final section: Do the authors have any plans to compare the reports made to the MeteoSwiss hail app with those reported by the hail disdrometers? Is there a particular reason why the crowd-sourced data were not used in this paper?

We agree with the referee that it would be extremely interesting to use both data sets in conjunction (hail sensors and crowdsourced reports). We are now thinking about the potential approaches that we could use to do so. We did not use the crowdsourced reports in

the present paper because we first wanted to focus on describing the observations of the automatic hail sensors. Moreover, most hail sensors are in areas with relatively low population density, which would limit the number of neighbouring crowdsourced reports.

Typographical Comments

We thank the referee for those comments; the text was corrected accordingly.

Page 2, line 38. Suggest saying, "Consequently, only a few papers in the scientific literature discuss local…."

Page 2, line 50: Say, "…how to make the best use of…".

Page 4, line 68: Suggest removing "from several events."

Page 4, line 81: Suggest replacing "masses" with "spheres."

Page 6, line 107: Suggest saying, "…slightly lower signal for the same hail size.".

Page 6, line 108: Suggest removing "…for analyzing a single hailstorm…"

Page 6, line 109: Replace "accuracy" with "precision."

Page 21, line 296: Suggest saying, "One reason for doing this is that…".

Page 21, lines 297-298: Suggest removing sentence starting with "Another." This text is redundant.